# *Debaryomyces hansenii*, *Stenotrophomonas rhizophila*, and Ulvan as Biocontrol Agents of Fruit Rot Disease in Muskmelon (*Cucumis melo* L.)

**DOI:** 10.3390/plants11020184

**Published:** 2022-01-11

**Authors:** Tomas Rivas-Garcia, Bernardo Murillo-Amador, Juan J. Reyes-Pérez, Roberto G. Chiquito-Contreras, Pablo Preciado-Rangel, Graciela D. Ávila-Quezada, Liliana Lara-Capistran, Luis G. Hernandez-Montiel

**Affiliations:** 1CONACYT-Universidad Autónoma Chapingo, Carretera Federal México-Texcoco km 38.5, San Diego, Texcoco 56230, Mexico; 2Centro de Investigaciones Biológicas del Noroeste, Calle Instituto Politécnico Nacional 195, Col. Playa Palo de Santa Rita Sur, La Paz 23096, Mexico; bmurillo04@cibnor.mx; 3Facultad de Ciencias Pecuarias, Universidad Técnica Estatal de Quevedo, Av. Quito km 1.5 vía a Santo Domingo, Quevedo 120501, Los Ríos, Ecuador; jreyes@uteq.edu.ec; 4Facultad de Ciencias Agrícolas, Universidad Veracruzana, Circuito Universitario Gonzalo Aguirre-Beltrán S/N, Zona Universitaria, Xalapa 91090, Mexico; rchiquito@uv.mx (R.G.C.-C.); lilara@uv.mx (L.L.-C.); 5Tecnológico Nacional de México, Instituto Tecnológico de Torreón, Carretera Torreón-San Pedro km 7.5, Ejido Ana, Torreón 27179, Mexico; pablo.pr@torreon.tecnm.mx; 6Facultad de Ciencias Agrotecnológicas, Universidad Autónoma de Chihuahua, Escorza 900, Col. Centro, Chihuahua 31000, Mexico; gdavila@uach.mx

**Keywords:** antioxidant enzymes, disease incidence, *Fusarium proliferatum*, mixed treatment

## Abstract

The indiscriminate use of synthetic fungicides has led to negative impact to human health and to the environment. Thus, we investigated the effects of postharvest biocontrol treatment with *Debaryomyces hansenii*, *Stenotrophomonas rhizophila*, and a polysaccharide ulvan on fruit rot disease, storability, and antioxidant enzyme activity in muskmelon (*Cucumis melo* L. var. *reticulatus*). Each fruit was treated with (1) 1 × 10^6^ cells mL^−1^ of *D. hansenii*, (2) 1 × 10^8^ CFU mL^−1^ of *S. rhizophila*, (3) 5 g L^−1^ of ulvan, (4) 1 × 10^6^ cells mL^−1^ of *D. hansenii +* 1 × 10^8^ CFU mL^−1^ of *S. rhizophila*, (5) 1 × 10^8^ CFU mL^−1^ of *S. rhizophila* + 5 g L^−1^ of ulvan, (6) 1 × 10^6^ cells mL^−1^ of *D. hansenii +* 1 × 10^8^ CFU mL^−1^ of *S. rhizophila* + 5 g L^−1^ of ulvan, (7) 1000 ppm of benomyl or sterile water (control). The fruits were air-dried for 2 h, and stored at 27 °C ± 1 °C and 85–90% relative humidity. The fruit rot disease was determined by estimating the disease incidence (%) and lesion diameter (mm), and the adhesion capacity of the biocontrol agents was observed via electron microscopy. Phytopathogen inoculation time before and after adding biocontrol agents were also recorded. Furthermore, the storability quality, weight loss (%), firmness (N), total soluble solids (%), and pH were quantified. The antioxidant enzymes including catalase, peroxidase, superoxide dismutase, and phenylalanine ammonium lyase were determined. In conclusion, the mixed treatment containing *D. hansenii*, *S. rhizophila*, and ulvan delayed fruit rot disease, preserved fruit quality, and increased antioxidant activity. The combined treatment is a promising and effective biological control method to promote the shelf life of harvested muskmelon.

## 1. Introduction

The muskmelon (*Cucumis melo* L.), belonging to the family Cucurbitaceae, is an important horticultural crop cultivated in temperate to arid regions in Asia (74%), America (11.9%), and Europe (7.2%), with a global production of 31,166,896 tons [1]. However, muskmelon is a climacteric ripening fruit, which deteriorates rapidly after harvesting because of pericarp browning and postharvest disease primarily induced by *Alternaria alternata*, *Rhizopus stolonifer*, *Trichothecium roseum*, and *Fusarium* spp. [2]. Postharvest fruit rot caused by *Fusarium* spp. is considered one of the main diseases that negatively impacts the quality, and influences the commercial acceptability and saleable stock of muskmelon [3]. Thus, muskmelon has a limited shelf life, which further limits their storage, transportation, and marketing [4]. Therefore, handling postharvest muskmelon, which is a key production concern, necessitates further research.

Many synthetic fungicides, such as acibenzolar-S-methyl, azoxystrobin, copper sulfate, imazalil, iprodione, and thiabendazole, are the most common commercial methods employed in muskmelon postharvest handling to retard fruit decay and prolong storage life [5,6]. Nonetheless, their indiscriminate use has led to residue accumulation in fruit, environmental pollution, carcinogenic risk to consumers, and pathogen resistance [7]. In addition, there is a trend to consume residue-free fruits, with stricter government regulations regarding agrochemical products [8]. Consequently, there is an essential need to find alternative methods such as biological control to inhibit decay in harvested fruit. Previous studies have shown that biological control by applying microbial antagonists, such as *Bacillus subtilis* [9], *Burkholderia* sp. [10], and *Pseudomonas graminis* [11], or by applying secondary metabolites such as phenylethyl alcohol from *Trichoderma asperellum* [12] and lactic acid from *Lactobacillus plantarum* [13], is a promising method for managing decay in harvested muskmelon.

Most microbial antagonists have been sourced from the fruit surface (epiphytic), but they can also be isolated from other nearby related areas, i.e., soil, roots, and the phyllosphere [14], or distant sources such as extremophile environments [15]. The marine yeast *Debaryomyces hansenii* has shown significant results as a biocontrol agent by diverse mechanisms of action, such as competition for space (i.e., inhibition of spore germination) and nutrients, and secondary metabolite excretion (i.e., volatile organic compounds and lytic enzymes) [16,17]. The marine bacteria *Stenotrophomonas rhizophila* have shown significant results as biocontrol agents by direct inhibition, excretion of volatile organic compounds, nutrient competition, and lytic enzymes [18,19]. Moreover, previous studies have demonstrated that *D. hansenii* and *S. rhizophila* are safe to humans [17,20].

However, microbial antagonists applied as a single treatment considerably vary in their efficiency and are inconsistent at high levels (>95%) of disease control than that of chemical fungicides [14]. Thus, the integrated approaches could be the key in the successful development of safe and sustainable alternatives for effective postharvest disease management in fruits [21]. Ulvan, a polysaccharide isolated from the green algae *Ulva* spp., has been demonstrated to induce resistance with no direct activity against other microorganisms such as *D. hansenii*, *S. rhizophila*, and *Fusarium proliferatum* [19]. However, the effects of individual or mixed postharvest treatment with *D. hansenii*, *S. rhizophila*, and ulvan on the quality and storability of harvested muskmelon have not been studied before.

In this study, the effects of *D. hansenii*, *S. rhizophila*, and ulvan as individual or mixed treatments on fruit rot disease, storage quality, and antioxidant enzyme activity in muskmelon (*Cucumis melo* L. var. *reticulatus*) was investigated. The aim of this study was to develop an effective and safe biological control strategy for inhibiting fruit decay and prolonging the shelf life of muskmelon.

## 2. Results

### 2.1. In Vivo Control Assay and Microscopic Visualization

The mixed treatment of *D. hansenii*, *S. rhizophila*, and ulvan significantly reduced the lesion diameter (3.5 mm) and significantly improved DC (73.5%) of fruit rot induced by *F. proliferatum* in muskmelon compared to that of the individual treatments, and observed better results than those treated with benomyl (Figure 1). Nevertheless, muskmelon fruit inoculated with only ulvan presented the highest lesion diameter (16.3 mm) and the lowest DC (14.3%). By applying Abbott’s formula, it was inferred that in comparison with single treatments, all the mixed treatments exhibited a synergistic effect on DC (Table 1). The mixed treatment with *D. hansenii*, *S. rhizophila*, and ulvan demonstrated the highest predicted synergistic effect.

Scanning electron micrograph imaging demonstrated that the spores and mycelia of *F. proliferatum* appeared and grew normally on muskmelon fruit in the control treatment (Figure 2a). When treated with biological control agents (BCAs) as a single treatment, *F. proliferatum* cells developed adhesion capacity (Figure 2b) with limited (Figure 2c) mycelial growth. In the mixed treatment with *D. hansenii* and *S. rhizophila*, the mycelial surface appeared abnormally shaped and notably damaged (Figure 2c).

### 2.2. Effect of Biocontrol Treatment Time on Their Biocontrol Efficacy

The effect of *D. hansenii*, *S. rhizophila*, and ulvan treatment time after or before the inoculation of *F. proliferatum* significantly affected DC (Table 2) and lesion diameter (Table 3). All muskmelon fruits treated before inoculating *F. proliferatum* had the highest DC and smaller lesion diameter than those treated after inoculating the phytopathogen. The longer the treatment time of the BCAs and ulvan before *F. proliferatum* inoculation, the higher the DC and the smaller the lesion diameter. The fruit inoculated with mixed treatment of *D. hansenii*, *S. rhizophila*, and ulvan 24 h before the inoculation of *F. proliferatum* presented the best results in DC (87.6%) and reduction in lesion diameter (1.7 mm). The fruits treated with ulvan 24 h after inoculating *F. proliferatum* had the lowest DC and the largest lesion diameter (27.7 mm). The DC and lesion diameter of muskmelon fruit treated with benomyl before or after *F. proliferatum* did not differ significantly. The results showed that *D. hansenii*, *S. rhizophila*, and ulvan are effective as preventive treatments rather than curative treatments.

### 2.3. Efficacy of Biocontrol Treatments on Natural Fruit Rot Development and Fruit Quality Parameters

Muskmelon fruits were dipped in either single or mixed treatments containing *D. hansenii*, *S. rhizophila*, and ulvan to assess natural fruit rot development and quality parameters. After 7 d of storage, DI significantly reduced with all treatments in comparison with the control treatment (70%) (Table 4). Muskmelon fruit immersed in the mixed treatment of BCAs and ulvan had the lowest DI (8.3%), which was even lower than that of benomyl (10%). All the mixed treatments had lower DI values than that of the single treatments. Regarding quality parameters, muskmelon fruit immersed in benomyl lost a significant amount of weight (0.68 g) and firmness (4.1 N) in comparison with those immersed in BCAs and ulvan as mixed or single treatments. Furthermore, TSS observed no significant difference between muskmelon treatments. Muskmelon fruit immersed in solutions of mixed treatments and a single treatment containing ulvan had the lowest pH values.

### 2.4. Antioxidant Enzymatic Activity on Muskmelon Fruit after Biocontrol Treatments

The antioxidant enzymatic activity in muskmelon was measured after treating with single or mixed solutions containing *D. hansenii*, *S. rhizophila*, and ulvan (Figure 3). SOD activity increased significantly in muskmelon fruit after 4 and 6 d of inoculation with the mixed treatment of BCAs and ulvan (Figure 3a), respectively. In all muskmelon fruits, SOD activity decreased considerably during the first 2 d of incubation and increased to the maximum activity level after 6 d of incubation. CAT activity in muskmelon fruits significantly decreased in all the treatments during the first 6 d of incubation and slightly increased after 8 d (Figure 3b). However, single and mixed treatments with *D. hansenii*, *S. rhizophila*, and ulvan maintained a higher CAT activity than that of the control treatment. The POX activity in muskmelon significantly increased with the single BCAs treatment after 6 d of inoculation in comparison with the control treatment (Figure 3c). The highest POX activity was quantified 4 d after inoculating muskmelon fruit with mixed treatment of *D. hansenii*, *S. rhizophila*, and ulvan. In all treatments, POX decreased gradually after incubating for 4 d. PAL activity significantly increased in all muskmelon fruits compared with that of the control treatment (Figure 3d). The highest PAL activity was quantified 2 d after inoculating the mixed treatment containing *D. hansenii*, *S. rhizophila*, and ulvan, which was maintained throughout the incubation period.

## 3. Discussion

Since publishing the first report on using *Bacillus subtilis* to treat brown rot caused by *Monilinia fructicola* on peaches in 1985, the use of microbial antagonists (i.e., yeast, bacteria, and fungi) as BCAs have been promoted as an alternative to chemical products [22]. Nonetheless, BCAs exhibit certain limitations because they are usually effective against specific hosts and well-defined phytopathogens, and are also affected by adverse environmental conditions [23]. Moreover, BCAs individually cannot eradicate established infections and cannot provide a broad-spectrum DC compared with that of chemical fungicides [24]. Additionally, BCAs must demonstrate a control efficiency comparable to that of conventional fungicides to be considered as a promising alternative [25]. Thus, combining BCAs with its compatible physical or chemical treatments is being investigated in recent years to enhance their individual performance through a synergistic effect [8]. Previous studies have developed several alternatives and compatible treatments, including physical treatments [26], resistance inducers [27], food additives [28], essential oils [29], low fungicidal doses [30], and mixed antagonist cultures [31].

In this study, the results demonstrated that mixed treatments containing BCAs and ulvan significantly enhanced the biocontrol effect of fruit rot disease in muskmelon compared with that of the single treatments. Mixed treatments with resistance inducers have been evaluated previously to enhance the activity of BCAs [27,32,33,34,35]. In a previous study, methyl jasmonate was inoculated as a mixed treatment to enhance the biocontrol effect of *Meyerozyma guilliermondii*, which reduced the disease incidence using the combined treatment (21%) in comparison with that of the individual yeast treatment (42%), which further affected the fungal morphology and upregulated resistance-related enzymes. The mixed treatments that include BCAs and resistance inducers are better than individual BCAs treatments because of their wide spectrum of action, and better efficiency for an expanded disease control under wide environmental conditions [36]. In this study, the compatible activity of the mixed treatment could be attributed to different ecological requirements of both BCAs [37], with ulvan not directly affecting these microorganisms [38].

Furthermore, the inoculation time of the BCAs in this study indicated that the reduced lesion size and DC are related to their high reproduction rate compared to that of the phytopathogen, which rapidly colonize the tissue during pre-treatment [37,39]. Ulvan inoculation time indicates that the reduction in lesion size and the decrease in disease are related to its ability to induce resistance and priming in fruits [40]. Therefore, the BCAs proposed in this study should be used in pretreatment to counteract melon fruit rot caused by *F. proliferatum*, Zhao et al. ref. [41] obtained similar results, wherein the efficiency of *Pichia guilliermondii* against *Rhizopus nigricans* was better when tomato fruits were treated 24 h before inoculating the phytopathogen. Besides, Lima et al. [42] reported that the combination of *Wickerhamomyces anomalus* and *Meyerozyma guilliermondii* inoculated 12 and 24 h before *Colletotrichum gloeosporioides* inoculation, reduced the disease incidence by 13.8% and 30%, respectively.

BCAs colonize more effectively the fruit host and limit the space and nutrients availability when they are inoculated before the phytopathogen (Figure 1). Thus, studying the effect of timing inoculation on the effectiveness of BCAs is essential to develop postharvest control strategies [14]. The time-related experiments in this study demonstrate the importance of applying BCAs immediately after fruit harvesting to control postharvest diseases and to preserve their quality parameters. The ability of BCAs as a preventive treatment rather than a corrective one is closely related to nutrient competition mechanisms [22,43,44,45].

The innate resistance to postharvest fungal decay is closely related to certain physiological parameters, such as senescence, which is remarkably decreased [46]. In a previous study, the effectiveness of *Pichia membranifaciens* as antagonist against *Penicillium expansum* in peach fruit could be enhanced by adding 0.2 g L^−1^ of benzo-(1,2,3)-thiadiazole-7-carbothioic acid *S*-methyl ester without reducing its quality parameters [47]. In this study, the mixed treatments of BCAs and ulvan significantly decreased the natural disease incidence, and preserved the firmness and weight of muskmelon. Initially, the TSS content in the fruit increased, probably due to the degradation of the non soluble polysaccharides to simple sugars, which later decreased with increase in storage time, and related to the increased respiration rate [48]. Furthermore, the respiration rate in muskmelon was delayed by the mixed treatment of BCAs and ulvan because of the increase in TSS post-treatment. The pH of muskmelon fruit decreased from an initial pH of 5.3 to 6.8 during fruit ripening [46]. Moreover, the enzyme polygalacturonase is associated with *F. proliferatum* pathogenicity and virulence, which acts more efficiently after an increase in pH during muskmelon fruit ripening [49,50]. In the results obtained, the mixed treatment with BCAs and ulvan had the lowest pH values, which could be associated with the lowest DI and lesion diameter according to the data presented previously in Section 2.3.

The efficiency of *D. hansenii*, *S. rhizophila*, and ulvan as single or mixed treatment(s) to control muskmelon fruit rot caused by *F. proliferatum* could be related to the increase in the defense response mechanism in fruit (i.e., priming, PR protein synthesis, and oxidative burst) [51]. *Debaryomyces hansenii* reportedly induces resistance in citrus fruits by increasing the synthesis of phytoalexins [52], which produce molecules that confer resistance in fruits against fungal phytopathogens [53]. The results obtained in this study are the first to report the induction of antioxidant enzymes in muskmelon by *S. rhizophila* to reduce the rot caused by *F. proliferatum*. Dumas et al. [54] determined that the defense induction in *Medicago truncatula* by ulvan is mediated by the jasmonate signaling pathway. In rice and wheat, ulvan induces priming and increases the first oxidative burst, increasing resistance against mildew [55]. Cluzet et al. [56] concluded that using microarrays helps ulvan increase the expression of genes coding for phytoalexins, PR proteins, and structural proteins. In this study, ulvan moderately affected the control of disease incidence; however, its effect is attributed to the induction of systemic acquired response (SAR) and priming mechanism, which operate after induced systemic response (ISR) [54,55,56]. However, elucidating the mechanisms involved in resistance induction in melon fruits by *D. hansenii*, *S. rhizophila*, and ulvan, requires further exhaustive investigation.

In previous reports, resistance induction was evidently promoted in melon fruits [57,58,59]. One of the initial defense responses against pathogens is the oxidative burst, which increased the reactive oxygen species (O^2−^ and H_2_O_2_) [60]. Although reactive oxygen species can contribute to defense in fruits, they can be degraded by antioxidant enzymes such as CAT, SOD, and POX [61]. CAT converts H_2_O_2_ to O_2_ and H_2_O, and POD degrades H_2_O_2_ by oxidizing phenolic compounds [62]. High levels of these enzymes are associated with reduced oxidative damage and delayed senescence [63]. PAL activity can be increased as part of the response mechanisms to numerous stress factors in the fruit [64]. According to Jetiyanon [65], the increase in PAL activity obtained by using the control can sufficiently inhibit pathogen invasion, and reduce disease incidence and lesion diameter [65].

## 4. Materials and Methods

### 4.1. Marine Microbial Antagonists Source and Concentration

*Debaryomyces hansenii* and *S. rhizophila* were obtained from the Phytopathology laboratory of Centro de Investigaciones Biologicas del Noroeste (CIBNOR), La Paz, Baja California Sur, Mexico. *Debaryomyces hansenii* and *S. rhizophila* were maintained and stored in potato dextrose agar (PDA; 39 g L^−1^) and trypticase soy agar (TSA, 40 g L^−1^) plates, respectively, at 4 °C. Liquid cultures of *D. hansenii* and *S. rhizophila* were prepared in 250 mL Erlenmeyer flasks containing 50 mL of potato dextrose broth (PDB, 39 g L^−1^) and trypticase soy broth (TSB, 40 g L^−1^), respectively, and were incubated at 27 °C for 24 h on a rotary shaker set at 180 rpm. *Debaryomyces hansenii* concentration was adjusted to 1 × 10^6^ cells mL^−1^ using a hemocytometer, and *S. rhizophila* concentration was adjusted to 1 × 10^8^ CFU mL^−1^ using a UV/Vis spectrophotometer (HACH, Dusseldorf, Germany) at 660 nm and absorbance of 1. *Debaryomyces hansenii* and *S. rhizophila* were adjusted to these concentrations prior to use in each of the following experiments.

### 4.2. Chemical Treatments Source and Concentration

Ulvan (OligoTech^®^, Elicityl Ltd., Crolles, France) solution was prepared by dissolving 5 g L^−1^ ulvan in sterile deionized water. The chemical fungicide benomyl was used at 1000 ppm. Ulvan and benomyl were adjusted to these concentrations prior to use in each of the following experiments.

### 4.3. Fusarium proliferatum Source and Concentration

*Fusarium proliferatum* was isolated from infected muskmelon fruit (*Cucumis melo* L. var. *reticulatus*) [38], and provided by CIBNOR. The fungus was stored on PDA at 4 °C. Prior to use, the culture was reactivated, and its pathogenicity was assessed by re-inoculating into wounded melon fruits, which was subsequently re-isolated onto PDA after establishing infection. Spore suspensions were obtained from 10-d old cultures maintained on PDA at 25 °C, and spore concentration was determined using a hemocytometer and adjusted to 1 × 10^4^ spores mL^−1^ with sterile distilled water (SDW) containing 0.05% (*v*/*v*) Tween 80. *Fusarium proliferatum* was adjusted to this concentration prior to use in each of the following experiments.

### 4.4. Muskmelon Fruit Source and Pre-Treatment

Muskmelon (*Cucumis melo* L. var. *reticulatus*) fruit were sampled at El Pescadero, Baja California Sur, Mexico from a commercial orchard. Fruits without mechanical injury, disease symptoms, physiological maturity, and of uniform size were chosen for the experiments. The fruit surface was disinfected with 1% sodium hypochlorite, washed with SDW, and air-dried at 27 °C.

### 4.5. In Vivo Biocontrol Assay and Microscopic Visualization

The biocontrol activity of *D. hansenii*, *S. rhizophila*, and ulvan was tested according to the method described by Zhang et al. [66]. Six equidistant wounds of 3-mm diameter were created on each fruit and inoculated with 20 µL of the following treatments: (1) *D. hansenii*, (2) *S. rhizophila*, (3) ulvan, (4) *D. hansenii* + *S. rhizophila*, (5) *D. hansenii* + ulvan, (6) *S. rhizophila* + ulvan, (7) *D. hansenii* + *S. rhizophila* + ulvan, and (8) benomyl. The fruits were dried for 2 h and then each wound was inoculated with 20 µL of a suspension adjusted of *F. proliferatum*. The treatments concentration was adjusted as described in Section 4.1, Section 4.2 and Section 4.3. Fruit were incubated at 27 °C and 90% relative humidity (RH) for 7 d. Disease control (DC) and lesion diameter (mm) were measured. The DC was calculated using the following formula:(1)$$\mathrm{DC}=100-\left[\left(\frac{\mathrm{Fi}}{\mathrm{Tf}}\right)\;\times \;100\right]$$
where Fi = is the number of infected fruits in each treatment, and Tf = is the total number of infected fruits in the control treatment.

The advantage of in vivo mixed biocontrol treatments were assessed with respect to the individual treatments (*D. hansenii*, *S. rhizophila*, and ulvan) and the type of interactions (additive, synergistic, or antagonistic). The synergy factor (SF) was calculated according to de Abbott’s formula [67]:(2)$$\mathrm{SF}=\frac{\mathrm{DC}}{\mathrm{DCE}}$$
where DC and DCE are the observed and expected disease control (%) of the mixed treatments, respectively. DCE was calculated using the following formula:(3)$$\mathrm{DCE}=\left(\mathrm{DCa}+\mathrm{DCb}+\mathrm{DCc}\right)-\left(\frac{\mathrm{DCa}\;\times \;\mathrm{DCb}\;\times \;\mathrm{DCc}}{100}\right)$$
where DCa, DCb, and DCc are the DCs of postharvest *D. hansenii*, *S. rhizophila*, and ulvan as single treatments, respectively.

For microscopic visualization, tissue samples of approximately 0.5 cm^2^ were collected from in vivo biocontrol assay and fixed as described by Rivas-Garcia et al. [37]. Samples were examined by scanning electron microscopy (SEM) (Hitachi^®^, S-3000 N, Tokyo, Japan). Each treatment was represented by five replicates with three fruits per replicate.

### 4.6. Effect of Biocontrol Treatment Time on the Control of Fruit Rot Disease

The in vivo effect of treatment time of *D. hansenii*, *S. rhizophila*, and ulvan on the suppression of *F. proliferatum* on muskmelon was assessed following the method described by Zhimo et al. [68] with some modifications. Muskmelon fruits were collected and prepared as described in Section 2.4 and inoculated with 20 µL suspensions of the following treatments: (1) *D. hansenii*, (2) *S. rhizophila*, (3) ulvan, (4) *D. hansenii* + *S. rhizophila*, (5) *D. hansenii* + ulvan, (6) *S. rhizophila* + ulvan, (7) *D. hansenii* + *S. rhizophila* + ulvan, and (8) benomyl either prior to (2, 12, and 24 h) or after (12 and 24 h) inoculating 20 µL of *F. proliferatum*. The treatments concentration was adjusted as described in Section 4.1, Section 4.2 and Section 4.3. The experiments were performed as previously described in Section 4.5. The fruits were dried for 2 h and then incubated at 27 °C and 90% RH for 7 d. The DC and lesion diameters (mm) were measured. Each treatment was represented by five replicates with three fruits per replicate.

### 4.7. Efficacy of Biocontrol Treatments on Natural Fruit Rot Development and Fruit Quality Parameters

Muskmelon fruits were collected, and immersed without a pre-treatment (Section 2.4) into 2 L plastic containers with the following treatments: (1) *D. hansenii*, (2) *S. rhizophila*, (3) ulvan, (4) *D. hansenii* + *S. rhizophila*, (5) *D. hansenii* + ulvan, (6) *S. rhizophila* + ulvan, (7) *D. hansenii* + *S. rhizophila* + ulvan, and (8) benomyl, for 2 min. The fruits were dried for 2 h and then each wound was inoculated with 20 µL of a suspension adjusted of *F. proliferatum*. The treatments concentration was adjusted as described in Section 4.1, Section 4.2 and Section 4.3. Fruits were incubated at 27 °C and 90% relative humidity (RH) for 7 d. The percentage of disease incidence (DI) was calculated using the formula:(4)$$\mathrm{DI}=\frac{\mathrm{Fi}}{\mathrm{Tf}}\times 100$$

The quality parameters measured in muskmelon included weight loss (%), fruit firmness (N), total soluble solids (%), and pH. For weight loss estimation, muskmelon fruit was weighed before and after storage. Firmness was measured by compressing the muskmelon fruit on two opposite sides along the equatorial region, after applying a load of 9.8 N with a texture analyzer. For total soluble solids (TSS) and pH, 10 g of muskmelon fruit was macerated to obtain fruit juice. TSS was determined using a digital Abbe refractometer (PR-32, Atago Co., Tokyo, Japan) at room temperature. The pH was measured using a digital pH meter (PHS-550; Lohand Co., Hnagzhou, China). Each treatment was represented by five replicates with three fruits per replicate.

### 4.8. Antioxidant Enzymatic Activity on Muskmelon Fruit

Muskmelon fruits were collected and prepared as described above (Section 2.4). Six equidistant wounds with 3-mm diameter in each fruit were inoculated with 20 µL of the following treatments: (1) *D. hansenii*, (2) *S. rhizophila*, (3) ulvan, (4) *D. hansenii* + *S. rhizophila*, (5) *D. hansenii* + ulvan, (6) *S. rhizophila* + ulvan, and (7) *D. hansenii* + *S. rhizophila* + ulvan. SDW was used as the control. The treatments concentration was adjusted as described in Section 4.1 and Section 4.2. The fruits were dried for 2 h, and incubated at 27 °C and 90% relative humidity (RH) for 8 d. Tissues adjacent to the inoculated area were sampled with a scalpel every 2 d (1 × 1 cm, length and depth), and were stored at −80 °C until enzymatic quantification. The collected samples were disrupted using liquid nitrogen and suspended in chilled phosphate buffer (0.1 M, pH 7.4) for catalase (CAT), peroxidase (POX), and superoxide dismutase (SOD) estimation, and suspended in chilled sodium borate buffer (0.1 M, pH 8) for phenylalanine ammonium lyase (PAL) quantification. The homogenate samples were centrifuged at 10,000× *g* for 20 min at 4 °C, and the supernatant was subjected to the enzymatic assay. CAT, POX, SOD, and PAL activities were measured using a commercial assay kit (Nanjing Jiancheng Bioengineering Institute, Nanjing, China).

Protein content was determined using the Bradford assay, with standard curve plotted using bovine serum albumin [69]. One unit of CAT activity is defined as the amount of enzyme that reacts with 1 nmol of formaldehyde per min and is expressed in min mg^−1^ of protein [70]. One unit of POX activity is defined as the amount of enzyme that causes the formation of tetra guaiacol in the presence of H_2_O_2_ per min and is expressed in U mg^−1^ of protein [71]. One unit of SOD activity is defined as the amount of enzyme necessary to inhibit 50% of the O_2_ reaction in the presence of nitro-blue tetrazolium reagent (NBT) and is expressed as U mg^−1^ of protein [72]. One unit of PAL activity is defined as µmol of cinnamic acid formed per minute per milligram of protein (min mg^−1^ of protein) [73]. Each treatment was represented by five replicates with three fruits per replicate.

### 4.9. Data Analysis

One-way analysis of variance (ANOVA) were performed to analyze the obtained data by using STATISTICA software (version 10.0; StatSoft, Tulsa, OK, USA). Post hoc least significant difference Fisher test (*p* ≤ 0.05) was used to compare means.

## 5. Conclusions

In this study, the mixed pre-treatment of *D. hansenii*, *S. rhizophila*, and ulvan enhanced the biocontrol effect on fruit rot disease in muskmelon, delayed natural fruit rot, lowered percentages of decay and weight loss, maintained higher antioxidant and defense-related enzymes (CAT, POX, SOD, and PAL), and preserved fruit quality (firmness, TSS, and pH). These results provide convincing evidence that postharvest treatment using 1 × 10^6^ cells of *D. hansenii*, 1 × 10^8^ cells of *S. rhizophila*, and ulvan displays higher disease resistance, better storability of harvested muskmelon fruit, and retains higher fruit quality, which suggests that postharvest mixed treatment containing BCAs and ulvan is a promising, safe, and effective biological control method in preserving the storage time of harvested muskmelon fruit. Omic technologies like metatranscriptomics and metagenomics analysis will be the future for the study of this complex tri-trophic interactions between microbial antagonists–fruit host–Pathogen.

## Figures and Tables

**Figure 1 plants-11-00184-f001:**
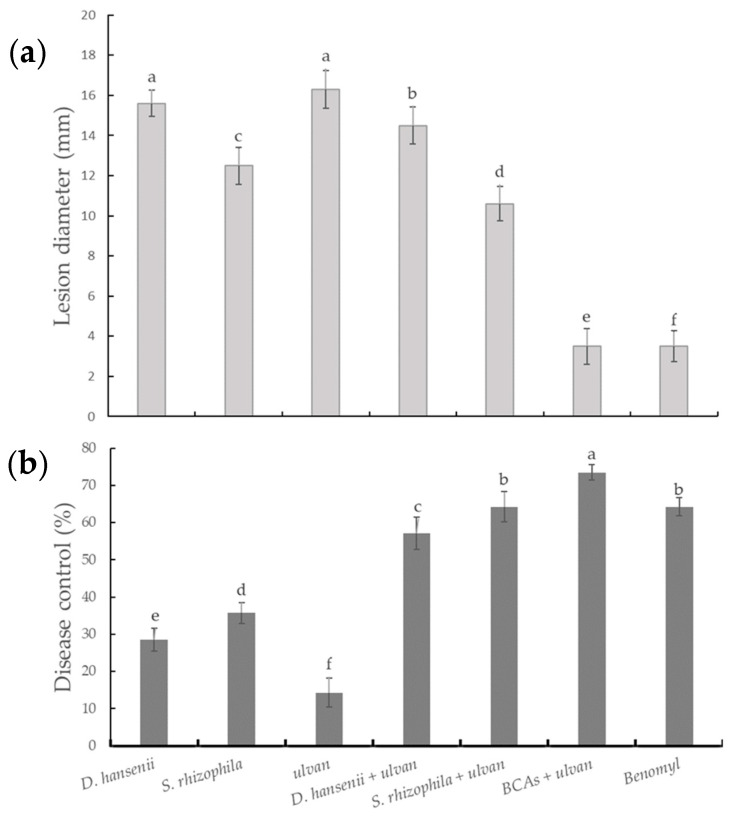
Effect of *D. hansenii*, *S. rhizophila,* and ulvan as individual or mixed treatments on the lesion diameter (**a**) and disease control (**b**) of fruit rot induced by *F. proliferatum* on muskmelon. Bars are the mean of five replicates with three fruits per replicate ± Standard deviation. Different letters in each column indicate significant difference (*p* ≤ 0.05). BCAs: Biological control agents (*D. hansenii* + *S. rhizophila*).

**Figure 2 plants-11-00184-f002:**
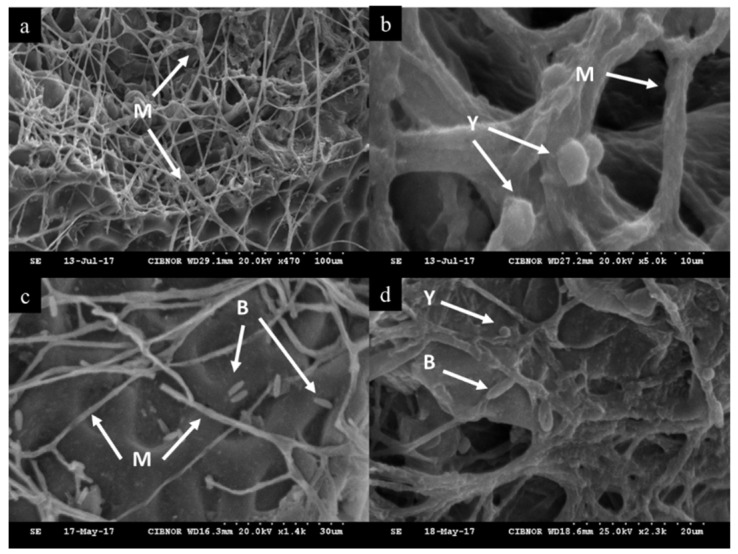
Scanning electron micrograph imaging of biocontrol activity of microbial antagonists against *F. proliferatum* in muskmelon fruit. (**a**) Mycelium of *F. proliferatum*; (**b**) Mycelium of *F. proliferatum* and cells of *D. hansenii*; (**c**) Mycelium of *F. proliferatum* and cells of *S. rhizophila*; (**d**) Mycelium of *F. proliferatum* and cells of *D. hansenii* and *S. rhizophila*. M = mycelium of *F. proliferatum*, Y = *D. hansenii*, and B = *S. rhizophila*.

**Figure 3 plants-11-00184-f003:**
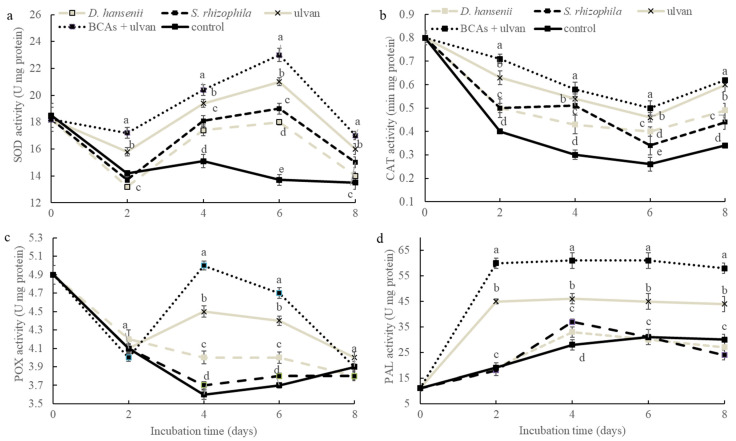
Effect of *D. hansenii*, *S. rhizophila* and ulvan on the antioxidant enzymatic activity on muskmelon fruit. BCAs = Biological control agents (*D. hansenii* + *S. rhizophila*). Each value is the mean of five replicates with three fruits per replicate ± Standard deviation. Different letters indicate significant difference (*p* ≤ 0.05). (**a**) superoxide dismutase (SOD) estimation; (**b**) catalase (CAT) estimation; (**c**) peroxidase (POX) estimation; (**d**) superoxide dismutase (SOD) estimation.

**Table 1 plants-11-00184-t001:** Synergistic biocontrol activity of *D. hansenii*, *S. rhizophila,* and ulvan against *F. proliferatum* on muskmelon fruit.

Treatment	DCE *	SF
*D. hansenii*	-	-
*S. rhizophila*	-	-
Ulvan	-	-
*D. hansenii* + *S. rhizophila*	49.2	1.7
*D. hansenii* + ulvan	38.4	1.5
*S. rhizophila* + ulvan	44.4	1.4
BCAs + ulvan	55.1	1.8
Benomyl	-	-

The disease control was calculated by comparing the treatments with the control (*F. proliferatum* + sterile distilled water) and was presented as percentage for each treatment. * DCE: Expected effect or expected control percentage, SF: Synergy factor, more information on materials and methods. BCAs: Biological control agents (*D. hansenii* + *S. rhizophila*).

**Table 2 plants-11-00184-t002:** Effect of inoculation timing of *D. hansenii*, *S. rhizophila*, and ulvan on disease control (%DC) by *F. proliferatum* on muskmelon fruit.

Treatment	Before (h)			After (h)	
	24	12	2	12	24
*D. hansenii*	39.7 ± 2.3 ^g,^*	35.2 ± 2.3 ^g^	28.6 ± 3.1 ^f^	17.2 ± 1.4 ^g^	10.7 ± 1.8 ^g^
*S. rhizophila*	44.3 ± 3.1 ^f^	40.2 ± 3.3 ^f^	35.7 ± 2.8 ^e^	20.6 ± 2.3 ^f^	13.6 ± 3.3 ^e^
Ulvan	48.9 ± 2.1 ^e^	27.4 ± 1.4 ^h^	14.3 ± 3.9 ^g^	7.2 ± 2.1 ^h^	5.3 ± 1.7 ^h^
BCAs	75.8 ± 1.1 ^b^	72.5 ± 1.8 ^b^	68.2 ± 3.5 ^b^	39.2 ± 3.8 ^c^	18.3 ± 3.4 ^b^
*D. hansenii* + ulvan	70.2 ± 1.3 ^c^	60.3 ± 2.2 ^e^	57.1 ± 4.3 ^d^	34.3 ± 1.6 ^d^	11.1 ± 2.1 ^f^
*S. rhizophila* + ulvan	75.3 ± 2.4 ^b^	68.8 ± 4.1 ^c^	64.3 ± 4.1 ^c^	25.7 ± 2.0 ^e^	15.7 ± 1.1 ^d^
BCAs + ulvan	87.6 ± 2.3 ^a^	80.7 ± 3.4 ^a^	73.5 ± 2.1 ^a^	40.1 ± 1.2 ^b^	17.4 ± 1.3 ^c^
Benomyl	66.5 ± 2.2 ^d^	65.7 ± 3.2 ^d^	64.3 ± 2.4 ^c^	60.5 ± 2.4 ^a^	59.5 ± 1.2 ^a^

The disease control was calculated by comparing the treatments with the control (*F. proliferatum* + sterile distilled water) and was expressed as percentage for each treatment. BCAs: Biological control agents (*D. hansenii* + *S. rhizophila*). * Each value is the mean of five replicates with three fruits per replicate ± Standard deviation. Different letters in each column indicate significant difference (*p* ≤ 0.05).

**Table 3 plants-11-00184-t003:** Effect of inoculation timing of *D. hansenii*, *S. rhizophila*, and ulvan on lesion diameter (mm) by *F. proliferatum* on muskmelon fruit.

Treatment	Before (h)			After (h)	
	24	12	2	12	24
*D. hansenii*	11.5 ± 0.5 ^b,^*	12.5 ± 0.7 ^b^	15.6 ± 0.7 ^c^	20.8 ± 0.9 ^c^	23.4 ± 0.7 ^d^
*S. rhizophila*	9.8 ± 0.6 ^d^	10.5 ± 0.8 ^c^	12.5 ± 0.9 ^e^	18.3 ± 0.5 ^e^	24.5 ± 1.3 ^c^
Ulvan	8.5 ± 0.3 ^e^	10.6 ± 0.6 ^c^	16.3 ± 0.9 ^b^	24.7 ± 1.1 ^b^	27.7 ± 0.8 ^b^
BCAs	7.8 ± 0.2 ^f^	8.7 ± 0.3 ^d^	10.6 ± 0.2 ^f^	15.3 ± 0.9 ^g^	17.1 ± 1.3 ^g^
*D. hansenii* + ulvan	10.2 ± 0.9 ^c^	12.4 ± 1.2 ^b^	14.5 ± 0.9 ^d^	19.4 ± 0.8 ^d^	20.6 ± 0.9 ^e^
*S. rhizophila* + ulvan	5.4 ± 0.2 ^h^	6.0 ± 0.3 ^e^	6.5 ± 0.9 ^g^	17.7 ± 1.1 ^f^	18.1 ± 1.2 ^f^
BCAs + ulvan	1.7 ± 0.2 ^i^	2.3 ± 0.1 ^f^	3.5 ± 0.2 ^h^	14.5 ± 0.3 ^h^	18.5 ± 0.1 ^f^
Benomyl	6.2 ± 0.3 ^g^	6.4 ± 0.2 ^e^	6.5 ± 0.7 ^g^	6.9 ± 0.5 ^i^	6.7 ± 0.3 ^h^
Control	27.2 ± 0.7 ^a^	27.8 ± 1.3 ^a^	28.4 ± 1.2 ^a^	28.8 ± 0.9 ^a^	29.1 ± 1.2 ^a^

BCAs: Biological control agents (*D. hansenii* + *S. rhizophila*). * Each value is the mean of five replicates with three fruits per replicate ± Standard deviation. Different letters in each column indicate significant difference (*p* ≤ 0.05).

**Table 4 plants-11-00184-t004:** Efficacy of *D. hansenii*, *S. rhizophila,* and ulvan on natural fruit rot development and fruit quality parameters.

Treatment	DI (%)	Weight Loss (g)	Firmness (N)	TSS (%)	pH
*D. hansenii*	33.3 ± 1.2 ^b,^*	0.30 ± 0.02 ^c^	4.2 ± 0.5 ^c^	9.2 ± 0.08 ^a^	6.5 ± 0.1 ^a^
*S. rhizophila*	26.7 ± 1.6 ^c^	0.30 ± 0.01 ^c^	4.2 ± 0.4 ^c^	9.2 ± 0.09 ^a^	6.5 ± 0.1 ^a^
Ulvan	23.3 ± 0.8 ^d^	0.24 ± 0.03 ^e^	4.2 ± 0.4 ^c^	9.3 ± 0.06 ^a^	6.1 ± 0.1 ^b^
BCAs	17.2 ± 1.1 ^f^	0.26 ± 0.02 ^d^	4.3 ± 0.3 ^a^	9.3 ± 0.04 ^a^	6.2 ± 0.2 ^b^
*D. hansenii* + ulvan	20.0 ± 1.2 ^e^	0.21 ± 0.03 ^f^	4.3 ± 0.3 ^a^	9.3 ± 0.08 ^a^	6.2 ± 0.1 ^b^
*S. rhizophila* + ulvan	13.3 ± 0.7 ^g^	0.22 ± 0.02 ^f^	4.3 ± 0.3 ^a^	9.3 ± 0.06 ^a^	6.1 ± 0.1 ^b^
BCAs + ulvan	8.3 ± 0.8 ^i^	0.21 ± 0.02 ^f^	4.3 ± 0.2 ^a^	9.3 ± 0.05 ^a^	6.0 ± 0.3 ^c^
Benomyl	10.0 ± 0.4 ^h^	0.68 ± 0.05 ^b^	4.1 ± 0.8 ^b^	9.2 ± 0.08 ^a^	6.6 ± 0.1 ^a^
Control	70.0 ± 1.4 ^a^	1.06 ± 0.08 ^a^	4.0 ± 0.6 ^d^	9.2 ± 0.07 ^a^	6.6 ± 0.1 ^a^

BCAs: Biological control agents (*D. hansenii* + *S. rhizophila*). * Each value is the mean of five replicates with three fruits per replicate ± Standard deviation. Different letters in each column indicate significant difference (*p* ≤ 0.05). DI means disease incidence, and TSS means total soluble solids.

## Data Availability

Not applicable.

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
