# Peer review of "Debaryomyces hansenii*, *Stenotrophomonas rhizophila*, and Ulvan as Biocontrol Agents of Fruit Rot Disease in Muskmelon (*Cucumis melo* L.)"

_plants, 2022, doi:10.3390/plants11020184_

Round 1

Reviewer 1 Report

This manuscript "Debaryomyces hansenii, Stenotrophomonas rhizophila, and Ulvan as Biocontrol Agents of Fruit Rot Disease in Muskmelon (Cucumis melo L.)" is an interesting piece of work worth publishing in Plants but it needs some revisions before it is accepted. The comments and suggestions are annotated in the manuscript. This can be accepted once the suggested revisions are properly done. 

Reviewer 2 Report

The revised manuscript presents interesting fungal biocontrol assay by combination of biological and chemical agents. This is very interesting and important topic for food producers.

I found this manuscript interesting, falling into scope of Plants Journal, well written and sound. Manuscript is well prepared with good English. The quality of figures is good. All tables and figures are informative and supported with text description and statistics.

The experimental design is very good, well-known and modern methods are used. The description and discussion are convincing allowing trust in the derived conclusions.

I found some issues as follows:

  1. There are some missing citations and/or cited with names not numbers - specifically in lines 199, 317, 342, 347
  2. In methods I miss some things:
  1. experiments have similar procedures (e.g. treatments) but there is no information about the cell densities nor concentrations of chemicals used (only volumes).
  2. only sections 4.6 and 4.8 mention number of fruits and treatment replications
  1. There are cell densities in the abstract, some data also are present in 'material preparation' section. It could be good to write a sub-section about the numbers of plants, treatment replications and cell densities/concentrations of agents used.

3.  I noticed that BCAs term is very often mentioned by its explanation is missing in methods and below Figure 1. In addition, I wonder about if in the Fig.1. caption 'F. proliferatum' is right. Shouldn't it be 'S. rhizophila'?

General comment: you have conducted statistics and may more often use 'significant' and 'non-significant' when describing results.

Significance was set for p≤0.05. It is not necessary to say every time it's Fisher's. In ANOVA Fisher statistic is used to derive p-value.

When citing Statistica please add USA after OK.

Very good piece of work worth to be published.
